# Current State and Future Challenges for PI3K Inhibitors in Cancer Therapy

**DOI:** 10.3390/cancers15030703

**Published:** 2023-01-23

**Authors:** Marianna Sirico, Alberto D’Angelo, Caterina Gianni, Chiara Casadei, Filippo Merloni, Ugo De Giorgi

**Affiliations:** 1Department of Medical Oncology, IRCCS Istituto Romagnolo per lo Studio dei Tumori (IRST) “Dino Amadori”, 47014 Meldola, Italy; 2Department of Life Sciences, University of Bath, Bath BA2 7AY, UK; 3Department of Oncology, Royal United Hospital, Bath BA1 3NG, UK

**Keywords:** PI3K inhibitors, mutations, clinical trial, target therapy

## Abstract

**Simple Summary:**

Phosphatidylinositol 3-kinase (PI3K) is a key regulator of many cellular processes and its hyperactivation promotes tumor cell growth and survival. A broad evaluation of the upstream and downstream nodes of its pathway allowed the discovery of several PI3K inhibitors (PI3Ki) with anti-tumor activity. However, the highly intrinsic toxicity and the onset of therapeutic resistance can limit their clinical application. To increase the antitumor effect and the therapeutic index, combination strategies and new dosing schedules have been investigated. However, further efforts are necessary to discover potentially actionable genetic alterations towards the goal of precision medicine.

**Abstract:**

The phosphoinositide 3 kinase (PI3K)-protein kinase B (PKB/AKT)-mammalian target of the rapamycin (mTOR) axis is a key signal transduction system that links oncogenes and multiple receptor classes which are involved in many essential cellular functions. Aberrant PI3K signalling is one of the most commonly mutated pathways in cancer. Consequently, more than 40 compounds targeting key components of this signalling network have been tested in clinical trials among various types of cancer. As the oncogenic activation of the PI3K/AKT/mTOR pathway often occurs alongside mutations in other signalling networks, combination therapy should be considered. In this review, we highlight recent advances in the knowledge of the PI3K pathway and discuss the current state and future challenges of targeting this pathway in clinical practice.

## 1. Introduction

Discovered in the late 1980s, the family of lipid kinases named phosphoinositide 3-kinase (PI3K) and the correlated PI3K/AKT signalling pathway have been shown to play a pivotal role in different oncogenic processes including cell survival, metabolism and metastasis [1]. The classical mechanisms behind the PI3K/AKT/mTOR pathway activation and its functions are described in Figure 1. Toward this goal, PI3K converts different signals from cytokines and growth factors into intracellular responses by producing phospholipids which, in turn, triggers the serine-threonine protein kinase AKT and downstream pathways [2]. While mTOR is one of the most common downstream effectors, the main critical regulator of the PI3K/AKT pathway is the phosphatase and tensin homologue (PTEN) tumour suppressor [3]. The PI3K/AKT pathway can be abnormally triggered in a wide range of cancers due to a plethora of mechanisms including somatic mutations and germline mutations in PIK3CA, AKT, PTEN and mTOR genes [4].

As a result, the PI3K/AKT pathway can be targeted by pharmacological molecules, thus making this pathway an interesting target for cancer intervention [5,6]. However, many issues regarding the use of pathway inhibitors, as well as the most effective drug to use in clinical practice, up to what cancer subtype might benefit the most from PI3K/Akt inhibitors, also due to the side effects, remain to be unsolved. Moreover, emerging evidence suggests that the PI3K/Akt pathway plays an immunomodulatory role [7]. In fact, several studies underlined how the PI3K pathway is involved in the differentiation of myeloid-derived suppressor cells (MDSCs) and Tregs into the tumor as well as the secretion of suppressive cytokines to impair stimulation of macrophages and dendritic cells, leading to an immunosuppressive tumour microenvironment (TME) [8]. This evidence suggests a potential synergy for combining PI3K inhibitors (PI3Kis) and immune-checkpoint inhibitors (ICIs).

In this review, we describe the critical role of the PI3K/AKT/mTOR pathway in tumorigenesis and the challenges in the clinical development of antitumour therapies targeting the PI3K/AKT/mTOR pathway, highlighting their limited clinical application. Finally, we provide an overview of the emerging data regarding PI3K/AKT/mTOR inhibitors in the most recent clinical trials, as well as their efficacy alone or in combinations for both solid and hematologic malignancies.

## 2. PI3K/AKT/mTOR Signalling in Cancer 

The PI3K/Akt/mTOR pathway has been associated with the development and progression of different neoplastic diseases [9]. For instance, almost 70% of breast [10] and ovarian cancers [11] carry an alteration of PI3K/AKT; similarly, the aberrant activation of the PI3K/AKT/mTOR pathway has been identified in 90% of lung adenocarcinomas (ADCs) and 40% of squamous cell carcinomas (SCCs) [12], leading to its hyperactivation.

Physiologically, the activity and homeostasis of the PI3K/AKT/mTOR pathway are strictly controlled by regulatory mechanisms; nevertheless, this pathway can be constitutively activated in several cancers. There are different mechanisms underlying this abnormal activatio including inactivating mutations in tumor suppressors genes such as PTEN or INPP4B, genomic alterations in PIK3CA, PIK3R1 (p85α regulatory subunit) or PIK3R2 (p85β regulatory subunit) and Akt subunits [13]. Mutations or overexpression of growth factor receptor (EGFR) or human growth factor receptor 2 (HER2), inactivating mutations in mTOR regulators gene such as TSC1 and TSC2, as well as the activating mutations in mTOR itself, are also detected across cancer types [14,15,16,17].

### 2.1. PI3Ks

The phosphoinositide 3-kinase (PI3K) family has an important role in a wide range of aspects of cell and tissue biology and a crucial role in human cancer [18]. The majority of PI3K functions are mediated by phosphoinositides, the low-abundance phosphorylated forms of phosphatidylinositol [19]. There are three different classes (I-II-II) of PIK3 according to their structural and specificity features. Class I PI3Ks are the most investigated and clinically interesting as they can be directly activated by cell surface receptors including G protein-coupled receptors (GPCRs), receptor tyrosine kinases (RTKs) and oncogenes such as G protein RAS [20].

#### 2.1.1. Class I

Activated as heterodimers, Class I PI3Ks are made of a regulatory (p85) and a catalytic subunit (p110), and they trigger downstream tyrosine kinases including GTPases and GPCRs [21]. It is worth mentioning that class I PI3Ks consist of four different catalytic isoforms (p110α, p110β, p110γ, p110δ) that are, respectively, expressed by PIK3CA, PIK3CB, PIK3CG and PIK3CD genes [22]. The most frequently mutated isoform in cancer is PIK3CA, whose mutations are an early event in colon and breast cancer [23]. While oncogenic mutations in PIK3CB are rare, reduced expression of PIK3CG expression has been associated with colon cancer development and progression [24]. On the contrary, PIK3CD is commonly found to be expressed in leukocytes and B cells, exerting a critical role in their growth and survival [25]. Of note, class I PI3Ks are further categorised into class IA and IB based on dissimilarity in the regulatory subunits. Several mechanisms are involved in the oncogenic activation of class IA PI3K including the inactivation of PTEN and p110 catalytic subunits, to mention a few [26].

#### 2.1.2. Class II

Activated as monomers, Class II PI3Ks consist of three catalytic components (C2α, C2β and C2γ) and no regulatory subunit [27]. They currently work as important signalling proteins with major roles under normal and pathological circumstances [28]. Indeed, while it has been demonstrated that PI3KC2α and PI3KC2β are widely expressed in the human body, the former plays a critical role in breast cancer invasiveness by impairing mitotic spindle formation [28,29]. Noteworthy, class II PI3Ks are engaged in the unique lipid molecule expression, with a critical function in cellular processes [30,31].

#### 2.1.3. Class III 

Class III PI3K VPS34 exerts its role in regulating macrophage phagocytosis and autophagy by connecting itself to a protein complex consisting of a catalytic and a regulatory subunit [22,30]. It has been shown that once activated, VPS34 is involved in transducing signals by modulating different protein kinases rather than directly regulating signalling pathways [32]. Indeed, emerging evidence has shown that VPS34 can modulate the basal activity of mTOR complex 1 (mTORC1) in animal models and the glycogen synthetase kinase 3 (GSK3) pathway in breast cancer patients treated with AKT inhibitors [33]. According to these observations, strategies targeting VPS34 might be an effective clinical treatment approach.

### 2.2. AKT

Also known as protein kinase B (PKB), the serine and threonine kinase AKT has been investigated since the early 90s and its dysfunction was observed in several diseases including cancer [34,35]. AKT1, AKT2 and AKT3 are the isoforms that have been identified so far and, although being found broadly expressed in the human body, they exert different and critical roles in cancer: for example, while AKT2 expression has been observed to increase in pancreatic cancer with a major role in cell migration and invasion, AKT3 expression was found increased in prostate and breast cancer [36]. Once triggered by receptor tyrosine kinases (RTKs), AKT can recruit and engage single or multiple subtypes of class I PI3Ks on the cell surface. In turn, activated PI3K enables PIP2 to PIP3 conversion and the consequent phosphoinositide-dependent-kinase-1 (PDK1) activation [37]. However, the regulation of AKT protein is a very complex system that includes other modulating factors such as IGF1, TRAF6 and TBK1 [38]. A new research thread is currently focusing on AKT in order to control class I PI3K signalling.

### 2.3. PTEN

Originally discovered as a mutated lipid phosphatase protein in a plethora of cancers, PTEN is now widely considered a tumour suppressor with a crucial role in the PI3K signalling pathway by preserving physiological cell activity [39]. It has been also observed that PTEN exerts its role in modulating the PI3K pathway by suppressing PIP2 to PIP3 conversion [40]. When PTEN is mutated or its function impaired, PI3K effectors, including AKT, become activated with no need for external oncogenic stimulus [41]. is normally involved in tumour signalling by dephosphorylation of targets including PTEN itself, insulin receptor substrate 1 (IRS1) and focal adhesion kinase (FAK) [42]. In cancers lacking PTEN function, the increased activation of AKT is a major oncogenic strategy [43]. It has also been demonstrated that PTEN is actively involved in angiogenesis and cancer cell migration [44].

### 2.4. mTOR

One of the main downstream targets of the PI3K/AKT pathway is mTOR, a protein kinase reported to regulate tumour development, metabolism, survival, angiogenesis and immunity [45]. When assembled in major complexes (mTORC1 and mTORC2), mTOR exerts significant roles in different physiological and pathological biological activities. While mTORC2 consists of mTOR, Rictor, SIN1 and mLST8 subunits, mTORC1 consists of mTOR, PRAS40, raptor and mLST8 subunits with a major role in modulating cell growth by phosphorylation of elF-4E-binding protein 1 (4EBP1) and S6 kinase 1 (S6K1) [46]. However, Akt was observed to have a close collaboration with mTOR via activation of the latter by phosphorylating the tuberous sclerosis complex 2 (TSC2) [47]. Once the mTORC2 is assembled, AKT is phosphorylated and activated by mTORC2 [48]. For this reason, mTOR inhibition has raised significant interest in clinical cancer research. According to experimental data and computer simulations, PI3K/AKT and MERK/ERK pathways can interact, activating or inhibiting each other with a context-dependent cross-talk [49]. Furthermore, different studies have shown that the blockade of one pathway may activate the other signalling cascade. For instance, PI3K inhibition induces the ERK2-dependent reactivation of AKT, eliminating the anti-clonogenic effect of inhibitors [50]. Therefore, the block of both MEK and PI3K/AKT/mTOR pathways with a combination of different signalling inhibitors may be used to more effectively target tumor cells, as compared with treatment with a single agent (Figure 1).

## 3. Targeting PI3K Pathway in Cancer

The PI3K/AKT/mTOR pathway is frequently activated in a wide variety of cancers including breast, gastric, ovarian, colorectal, prostate, glioblastoma and endometrial cancers [27]. In addition, it plays a key role in cell survival, proliferation, differentiation and glucose transport [13], and its hyperactivation can induce resistance to antitumor treatment [51,52]. As a result, the PI3K pathway and associated components have become an attractive anticancer drug target.

### 3.1. PI3K Inhibitors 

Recently, a plethora of PI3Kis have been investigated in clinical trials and subsequently approved as potential chemotherapeutic drugs for cancer therapy. PI3Kis are grouped into pan-PI3K inhibitors, isoform-selective PI3K inhibitors and dual PI3K/mTOR.

#### 3.1.1. Pan-PI3K Inhibitors 

The first generation of PI3Kis, defined as pan-PI3Ki, target all four catalytic isoforms of class I PI3Ks (α, β, γ, and δ) [53,54] and includes pictilisib (GDC-0941), buparlisib (BKM120) and copalinsib. These small molecules have shown a broad spectrum of activities, as well as a broader inhibition, leading to severe adverse events and treatment discontinuation [55]. Despite their high toxicity, multiple efforts have been carried out to develop agents accurately targeting PI3K isoforms to improve the therapeutic outcome.

Pictilisib (GDC-0941), a thienopyrimidine molecule, is a robust, selective, orally available PI3K inhibitor and the first PI3Ki to be assessed in a clinical trial. This drug demonstrated solid efficacy in human tumor xenografts murine models of U87MG glioblastoma and IGROV1 ovarian cancer, alone or in combination with other targeted therapy [20,56]. In a phase 1 dose-escalation clinical trial, a partial response was observed in a patient with V600E BRAF-mutant melanoma and in a patient with platinum-refractory epithelial ovarian cancer, exhibiting PTEN loss and PIK3CA amplification [57]. The most common toxicities were low-grade nausea, rash and fatigue, with one patient reporting grade 3 hyperglycemia [57].

Buparlisib (BKM120) is another potent and selective pan-class I PI3Ki which can cross the blood–brain barrier and potentially lead to PI3K inhibition in the brain [58]. Speranza et al. have found out that buparlisib has potent anti-invasive effects in glioblastoma cell lines and in in vitro patient-derived glioma cells, with no significant adverse effects [59]. As a consequence of PI3K inhibition in the central nervous system, a small number of patients experienced mood alterations such as anxiety, irritability or depression, which are generally mild and responsive to dose reductions [58,60].

In a very small subset of triple-negative breast cancer (TNBC) patients treated with buparlisib, Garrido-Castro et al. observed a prolonged stable disease (SD), although an objective response was not confirmed [61]. Given the emerging evidence regarding acquired endocrine resistance and PI3K activation [62,63], buparlisib has been evaluated in hormone receptor-positive/HER-2 negative metastatic breast cancer (HR+/HER2− MBC). In a phase 1b trial, Mayer et al. showed that the combination of buparlisib plus letrozole was safe with reversible toxicity in HR+/HER2− MBC refractory to endocrine therapy (ET) [60]. In the same setting, the BELLE 3 trial demonstrated that buparlisib plus fulvestrant was associated with better progression-free survival (PFS) compared to fulvestrant alone (median 3.9 months versus 1.8 months; HR 0.67 CI 0.53–0.84; *p* < 0.001), especially in patients with a real-time polymerase chain reaction (RT-PCR) or ctDNA PIK3CA mutations [64]. Alongside the positive findings with endocrine therapy, the addition of buparlisib to chemotherapy did not improve the median PFS [65]. Unfortunately, buparlisib administration was limited due to the high metabolic and psychiatric toxicity that emerged from clinical trials leading to the discontinuation of drug development. Additional data are required to confirm the clinical use of buparlisib.

Copanlisib is a potent, highly selective, pan-class I PI3Ki with a predominant activity against the isoforms p110α and p110δ [66]. It is administered intravenously while the other PI3K inhibitors, such as idelalisib and duvelisib, are normally administered orally [66]. Copanlisib was approved by FDA for the treatment of adult patients with relapsed follicular lymphoma (FL) who received at least two prior systemic therapies [67]. Interestingly, copanlisib is being studied in advanced HER2+ BC in addition to pertuzumab and trastuzumab to evaluate if this combination can overcome the resistance caused by the hyperactivation of the PI3K pathway [68]. Its use is associated with hypertension, diarrhoea and transient hyperglycemia, which is a common and predictable effect of PI3Kα inhibition due to the abrogation of downstream insulin receptor signalling [69].

#### 3.1.2. Isoform-Selective PI3K Inhibitors 

Although they require a stricter selection of patients, isoform-selective PI3K inhibitors are characterised by improved efficacy and fewer adverse events (AEs) compared to pan-PI3Ki [70]. The safer profile allowed isoform-selective PI3Ki to be developed and approved for clinical practice.

Alpelisib (BYL719) is the first oral isoform-selective PI3Ki targeting the p110α isoform of wild-type PI3Kα to be approved by the US Food and Drug Administration (FDA) and by the European Medicines Agency (EMA) [71]. Compared to the other isoform, alpelisib specificity induces a 50 times stronger activity against PI3Kα [72,73]. In 2019, Juric et al. conducted a phase 1b trial to assess the maximum tolerated dose of alpelisib (MTD) in patients with HR+/HER2− BC [74]. They observed that alpelisib plus fulvestrant lead to an improvement in PFS and OS in patients with PI3KCA alterations compared to the wild-type group, with manageable toxicity [74]. Similarly, Andrè et al. conducted a phase 3 clinical trial of alpelisib in combination with fulvestrant for the treatment of MBC [75]. They observed an increased PFS (7.4 months versus 5.6 months; HR: 0.85 95% CI 0.58–1.25) and objective response (OR) in patients treated with alpelisib and fulvestrant compared to the control arm (26% versus 23.8%). Nevertheless, patients in the experimental arm experienced a higher rate of hyperglycemia, rashes and diarrhoea compared to the placebo arm [75]. Following the aforementioned promising outcomes, alpelisib was approved by FDA in 2019 for the treatment of PIK3CA-mutant, HR+/HER2− MBC [65]. It is worthwhile to point out that alpelisib was approved in the USA with the companion diagnostic test Therascreen^®^ PIK3CA RGQ PCR kit (Qiagen, Hilden, Germany).

In 2022, Rugo et al., using comprehensive genomic profiling (CGP), detected approximately 72% PIK3CA mutations (PIK3Cm) in tissue biopsies from 33,977 patients with MBC and demonstrated that up to 20% of patients carried PIK3CA mutations which have not been identified by Therascreen® PIK3CA [71]. Of note, this study found out that these patients with different PI3KCAm also have longer PFS when administered with alpelisib plus fulvestrant compared to fulvestrant alone [71].

So far, the optimal method to detect PIKCAm in clinical practice is not yet established, and prospective clinical trials are warranted to demonstrate the PI3K inhibitors benefit in patients with PI3KCA mutations, not included in the SOLAR-1 trial.

Taselisib (GDC-0032) is a novel potent inhibitor of PI3Kα, exerting its blocking activity onp110ɑ, p110γ and p110δ proteins. In a preclinical study, taselisib showed significant antiproliferative activity in head and neck squamous carcinomas (HNSCC) cell lines harbouring PIK3CA-activating mutations [76]. Moreover, in the same setting, the combination of taselisib and radiotherapy was more efficacious than treatment alone both in vitro and in vivo [76]. Following preclinical studies, taselisib showed clinical activity in a phase I dose-finding clinical trial in patients with PIK3CA-mutant solid tumors, especially in MBC with a 36% of overall response rate (ORR) [74]. Consequently, Baselga et al. evaluated the efficacy of taselisib and fulvestrant in a phase 3 trial for HR+/PI3KCA-mutated MBC patients. The study reported a modest PFS increase in the efficacy of the combination treatment compared to fulvestrant alone (median PFS 7.4 months versus 5.4 months; HR 0.70, *p* < 0.01). However, the modest PFS improvement was associated with significant toxicity, especially diarrhoea (grade 3/4 of 12% vs. <1% for hormonal therapy alone) and hyperglycemia (grade 3/4 of 11% for the taselisib arm vs. <1% for the control arm), resulting in discontinuation of drug development for this subgroup [77,78].

The greater selectivity for the mutant PI3Kɑ isoform and the stronger inhibitory effect may justify why taselisib is correlated with a worse toxicity profile compared to alpelisib [44].

Idelalisib (Zydelig) is an orally bioavailable ATP-competitive kinase inhibitor specifically designed to target the phosphoinositide 3-kinase p110 isoform δ (PI3Kδ) with accurate selectivity and potency [76]. Due to its hyperactivation in B-cell malignancies and its crucial role in the B-cell receptor (BCR) pathway, it has been approved by FDA in 2014 for the treatment of indolent B-cell malignancies including relapsed/refractory chronic lymphocytic leukaemia (CLL), in association with rituximab, as monotherapy for relapsed follicular lymphoma (FL) and relapsed small lymphocytic leukaemia (SLL), in patients who received at least two prior systemic therapies [79]. Several clinical trials are ongoing to determine the activity, efficacy, and toxicity profile of PIK3CA inhibitors alone or in combination (Table 1).

### 3.2. AKT Inhibitors

Due to its role as a key-molecular regulator of the PI3K/AKT/mTOR pathway, AKT could be an interesting target. Indeed, AKT inhibition induces the block of mTORC1 activation, leading to the control of the downstream effects of the PI3K/AKT/mTOR cascade [11,77].

The majority of AKT inhibitors investigated so far in clinical trials can inhibit all three AKT subunits and, for this reason, they are defined as pan-AKT inhibitors. Several Akt-inhibitors, such as MK2206, capivasertib (AZD5363), afuresertib (GSK2110183) and ipatasertib have been developed to target AKT signalling in vitro and in vivo; however, none of them has yet received FDA approval for cancer treatment.

MK220 is a first-in-class allosteric AKT1/2/3 inhibitor with evidence of preclinical efficacy when combined with cytotoxic agents including doxorubicin, gemcitabine, docetaxel and carboplatin in the lung NCI-H460 cell line [41].

In preclinical studies, this drug has been demonstrated to restore erlotinib activity in erlotinib-sensitive and resistant non-small cell lung cancer (NSCLC) cell lines [80]. Additionally, it showed encouraging anti-tumour activity in acute myeloid leukaemia (AML) [81] and an ability to inhibit both AKT and mTOR signalling in nasopharyngeal carcinoma (NPC) cell lines [82]. Moreover, MK220 has shown preliminary activity in different phase I trials [81,83], and it is being currently tested in phase II trials as a monotherapy in metastatic pancreatic cancer [84] or in combination with the MEK inhibitor selumetinib (+MK2206) in colon-rectal cancer [84].

Capivasertib (AZD5363) is a novel, selective ATP-competitive pan-AKT kinase inhibitor that exerts activity against the three AKT isoforms (AKT1, AKT2 and AKT3) [85]. It can potentially treat a wide range of solid and hematologic malignancies as a monotherapy or in combination, both in vivo and in vitro [86]. Tumor types carrying PTEN mutation, PIK3CA mutation, or HER2 amplification, without coincident RAS mutation, are strongly associated with preclinical sensitivity to capivasertib [87]. In preclinical BC models, capivasertib can overcome resistance or increase sensitivity to HER2 inhibitors and improve chemotherapy efficacy, leading to tumor regression [87]. Similarly, capivasertib as a monotherapy or combined with different drugs has demonstrated preclinical efficacy in castrate-resistant prostate cancer (CRPC) [88], PI3KCA-mutant gastric cancer [89], trastuzumab-resistant esophagal squamous cell carcinoma [90] and NSCLC [91]. Furthermore, in 2020, Smith et al. demonstrated that capivasertib alone or in combination with fulvestrant was well tolerated and showed promising anticancer activity in patients with AKT^1E17K-mutant^ HR+/MBC in a phase I expansion study [92]. In the same setting, the phase 2 randomized FAKTION trial, demonstrated that the addition of capivasertib to fulvestrant resulted in a significantly longer PFS [93]. At the 2022 San Antonio Breast Cancer Symposium (SABCS), Turner presented the results from the CAPItello-291 phase III trial, showing a statistically significant and clinically meaningful improvement in PFS with the combination of fulvestrant and capivasertib in patients with HER2+/HER2-low or negative MBC, following recurrence or progression on, or after, endocrine therapy (with or without a CDK4/6 inhibitor) [94].

Given the synergy between poly(ADP-ribose) polymerase (PARP) and PI3Ki in preclinical data, Trap et al., evaluated capivasertib and the PARP inhibitor (PARPi) olaparib in a phase 1 study [91]. They observed that capivasertib was safe and well tolerated. Furthermore, antitumor activity was reported in both patients harbouring germline BRCA1/2 mutations and BRCA1/2 wild-type and with or without the somatic DNA damage repair gene (DDR) and/or PI3K/AKT pathway alterations [95].

It is worth mentioning that large-scale genomic studies of human cancer demonstrated that AKT1-E17K is the most common AKT mutation and improves the efficacy of AKT inhibitor therapy in solid tumors [87,96]. In a multicohort basket study, capivasertib obtained promising PFS outcomes in heavily pretreated AKT1 E17K-mutant breast and gynecologic cancer patients [96]. However, the response rate was lower than the response to those therapies targeting EGFR, ALK, ROS1 and BRAF. As a consequence, the full potential of capivasertib in AKT1-mutant cancers may require drug combination.

The large phase 2 screening trial MATCH (Molecular Analysis for Therapy Choice) (NCT02465060) is ongoing to match targeted therapy in 6452 patients with solid tumors or lymphomas, harbouring specific mutations, that have progressed to first-line standard treatment. Capivasertib and ipatasertib are eligible for patients with AKT mutations, while the PI3K inhibitor GSK2636771 is indicated for patients with PTEN mutation or deletion. On the other hand, the MyTACTIC trial is a phase II, multi-arm study investigating the safety and efficacy of targeted therapies in unresectable or metastatic solid tumors harbouring genomic alterations or protein expression patterns, predictive of response. This trial includes ipatasertib for patients with AKT1/2/3 mutations or PTEN loss of function and inavolisib for patients with PIK3CA mutations.

Afuresertib is another ATP-competitive AKT inhibitor that has been investigated in a phase Ib/II dose escalation study in combination with carboplatin and paclitaxel in recurrent platinum-resistant ovarian cancer. The study reported an ORR of 32% and a median PFS of 7.1 months [97]. Ultimately, ipatasertib is an additional highly selective oral ATP-competitive pan-AKT inhibitor showing encouraging activity, especially in tumors with markers of AKT activation, including high-basal phospho-AKT levels, PTEN loss and PIK3CA kinase domain mutations [98,99]. In a phase 1b trial, ipatasertib in combination with chemotherapy or hormone therapy was well tolerated and demonstrated radiographic responses in patients with MBC with a safe toxicity profile [100]. Similarly, the phase 2 randomized LOTUS trial showed an improvement in PFS with the addition of ipatasertib to paclitaxel in TNBC [101]. Conversely, in the phase 3 IPATunity130 trial, the addition of ipatasertib to paclitaxel did not improve efficacy in PIK3CA/AKT/PTEN-altered HR+/HER2− MBC [102]. These findings are consistent with the results of the BEECH trial where the combination of paclitaxel and an AKT inhibitor did not improve the PFS neither in the overall population nor in the PIK3CA-altered population [103]. A possible explanation can be the higher number of patients discontinuing paclitaxel due to ipatasertib adverse events (AEs) [102]. From the SOLAR1 and the FAKTION trial, it appears that endocrine blockade may be essential in order to obtain greater clinical benefit from AKT inhibition in HR+/HER2− MBC.

Different clinical trials are ongoing to assess the activity, efficacy, and toxicity profile of PIK3i alone or in combination with other target therapies (Table 2).

### 3.3. mTORC1 and mTORC2 Inhibitors

mTOR is a protein kinase which is extensively associated with cell growth, metabolism, survival, catabolism and autophagy [94], and is observed hyperactive in 40 to 90% of solid tumors [95]. mTOR is a downstream effector of the PI3K oncogenic pathway and it is the main reason behind the development of catalytic domain inhibitors, capable of blocking both mTOR and PI3K [96]. Rapamycin and its analogue (everolimus, temsirolimus and deforolimus) represent the first generation of mTOR inhibitors (mTORi), which are able to selectively inhibit the mTORC1 activity [97]. Rapamycin is a natural product that inhibits mTOR with high specificity [98]; however, its clinical application was limited due to its poor solubility and stability [97]. Therefore, rapamycin analogues with better solubility and metabolic properties have been developed. Water-soluble temsirolimus and deforolimus can be administered intravenously, while rapamycin and everolimus have lower solubility and can be administered orally [99].

In addition, rapamycin dosage may also affect mTOR activity [104]. Indeed, low nanomolar doses of rapamycin can impair S6K phosphorylation by mTORC1, delaying G1 cell-cycle progression [105,106]. On the other hand, micromolar doses of rapamycin might suppress the phosphorylation of both S6K and 4E-BP1 [107,108]. Unfortunately, this treatment can frequently result in a feedback activation of AKT phosphorylation by mTORC2 [109,110], which promotes cell survival [104,111].

#### 3.3.1. ATP-Competitive mTOR Inhibitors 

In order to more efficiently inhibit mTOR, a second generation of mTOR inhibitors targeting both mTORC1 and mTORC2 have been developed, also called selective mTOR kinase inhibitors (TORKIs) [112]. These small molecules, classified as ATP analogues, provide a robust inhibition of both mTORC1/2, and can reduce the resistance observed with rapamycin analogues [112]. Although ATP analogues showed a higher inhibitory effect in preclinical studies [113], large clinical trials have not been conducted yet and TORKIs are still not approved for clinical use.

#### 3.3.2. Dual PI3K/mTOR Inhibitors

Even though the inhibition of mTORC1 and mTORC2 can lead to a downregulation of AKT S473 phosphorylation, mTOR inhibition may paradoxically induce the activation of the PI3K/PDK1 axis. Therefore, the inhibition of both PI3K and mTOR may enhance anti-tumor activity compared to the mTOR-block alone [114,115].

Dual PI3K/mTOR inhibitors (PI3K/mTORi) include SF1126, dactolisib (BEZ235), voxtalisib (XL765) and gedas (PKI-587) [57]. Dactolisib elicited antitumor activity in human glioblastoma (GBM) cell lines and an orthotopic xenograft model [101], whereas voxtalisib showed encouraging efficacy with an acceptable safety profile in patients with follicular lymphoma [102]. Moreover, in T-cell acute lymphoblastic leukaemia (T-ALL), the dual-specificity of PI3K/mTORi PKI-587 was the most selective for T-ALL cells dependent on the PI3K/mTOR pathway [103]. Finally, this class of drugs has the potential to treat tumors with a wide range of genetic abnormalities including PTEN and TSC1/2 loss of function and STK11 alterations [35], with the latter being found in a third of NSCLC and associated with KRAS mutations [116]. Additionally, they exhibit a broad activity profile and significantly higher toxicity [117] (Figure 2).

Several trials are currently ongoing to establish the efficacy of dual PI3K/mTORi (Table 3).

### 3.4. Combination Strategies 

Acquired and intrinsic drug resistance with monotherapy is a major limit to PI3K inhibitors efficacy and it may be attributed to the complex feedback in the PI3K/AKT/mTOR signalling and its crosstalk with other pathways. Considering the well-establish evidence from preclinical studies, potential drug combinations may include chemotherapy, kinase inhibitors and ICIs.

#### 3.4.1. Her-2 Inhibitors

Aberrant activation of the ErbB family of receptors is one of the most common causes of cancer [118]. EGFR and HER2 are members of the ErbB family of RTKs and they have a crucial role in cell proliferation and survival [119]. For instance, an important group of studies showed that her2/neu gene amplification is common in human BC and it is correlated with poor prognosis [120]. To date, targeting the HER2-receptor has significantly changed cancer therapy, preventing signal initiation and crosstalk with complementary pathways but also improving the sensitivity of tumor cells to both chemotherapy and radiation [121]. In a phase 1b trial, Zambrano et al. observed that buparlisib might be combined with paclitaxel trastuzumab in HER2+ MBC [122]. Similarly, Pistilli et al. showed that buparlisib plus trastuzumab regimen has an acceptable safety profile but limited efficacy in patients with heavily pretreated and trastuzumab-resistant HER2+ MBC, and patients with progressive brain metastases also receiving capecitabine [120].

#### 3.4.2. MAPK Inhibitors 

It is well established that PI3K/AKT/mTOR and RAS/RAF/MEK/ERK pathways interact with each other at several nodes, leading to a potential pathway convergence for the development of drug combinations [123]. Indeed, parallel activation of the PI3K/AKT/mTOR pathway may be responsible for primary and acquired resistance to BRAF-targeted therapy [124]. The results of a phase 1b trial by Shapiro et al. showed that the MEK inhibitors cobimetinib and pictilisib had limited tolerability and efficacy in solid tumors [125]. Increasing evidence suggests that dual blockade of both pathways has a critical role in tumors with a high frequency of RAS/RAF/MEK/ERK pathway activation and when double blockade is required to overcome drug resistance [126].

More specifically, melanoma and BC frequently exhibited hyperactivation of PI3K and PI3K/AKT/mTOR and MAPK/MEK/ERK pathways [125,127]. Numerous preclinical studies have demonstrated that dual pharmacological inhibition of PI3K and MAPK pathways (via both continuous and intermittent dosing) increased therapeutic efficacy in basal-like BC and melanoma models [128,129]. As a consequence, a phase Ib study has been conducted to test the MEK inhibitors pimasertib combined with voxtalisib in patients with advanced solid tumors, including TNBC and BRAFV600-mutant melanoma, who progressed on BRAF inhibitors [130]. However, the combination showed poor long-term tolerability and limited anti-tumour activity, preventing it from progressing into further testing [130]. Similar dose-limiting toxicities emerged in BRAFV600-mutant advanced melanoma patients treated with the combination of buparlisib and the BRAF inhibitor vemurafenib [131]. Regarding AKT inhibitors, uprosertib in combination with the oral MEK1/MEK2 inhibitor trametinib showed poor tolerability in patients with solid tumors and minimal clinical efficacy [132]. Similarly, the trametinib and afuresertib combination was poorly tolerated in patients with solid tumors and multiple myeloma [117]. It is worth mentioning that both RAS/MAPK and PI3K pathways play a key role in cancer metabolism. PI3K can directly reset cellular metabolism by phosphorylating metabolic enzymes and regulating metabolism-associated proteins such as sterol regulatory element-binding proteins (SREBP), thus enhancing the activities of nutrient transporters indirectly by controlling various transcriptional factors (TFs) [24,133].

Similarly, RAS/MAPK signalling is involved in glucose metabolism in different ways. Mutant KRAS upregulates the hexokinase 1 and 2 (HK1 and HK2) rate-limiting enzymes of glycolysis [134,135] and increases the expression of key glycolytic enzymes such as PFK1, ENO1, and LDHA [136], thus stimulating glycolytic flux and facilitating the synthesis of glycolytic intermediates [137,138,139].

Given the interest in combination treatment of MAPK and PI3K/AKT pathway inhibitors, further investigation may be warranted, especially in patients with coexisting PI3K pathway mutations and KRAS or BRAF mutations.

#### 3.4.3. Chemotherapy 

It is well-established that the PI3K pathway synergizes with various chemotherapeutic agents such as doxorubicin, etoposide, topotecan, cisplatin, vincristine and taxol, resulting in increased tumour sensitivity to chemotherapy [140]. Interestingly enough, PI3K inhibition was reported to induce apoptosis and suppress tumor growth in patients’ derived primary neuroblastoma cells and in an in vivo neuroblastoma model [141]. Additionally, preliminary clinical studies demonstrated that PI3Ki in combination with chemotherapy are safe and well tolerated [27]. Pictilisib, carboplatin and paclitaxel have demonstrated promising antitumor activity in patients with NSCLC [142]. In terms of clinical benefit, the addition of ipatasertib to mFOLFOX6 did not improve PFS in a phase 2 randomized trial enrolling metastatic gastric or gastroesophageal junction cancer patients [143]. According to preclinical models, PI3K signalling stabilizes and preserves DNA double-strand break (DSB) repair by interacting with the homologous recombination (HR) complex [144], and is fundamental for DNA repair during ionizing radiation [145]. It is jointly agreed that PI3K inhibition may induce DNA damage and subsequently increase the sensitivity of cell lines to PARPi [146,147]. Given this evidence, Ibrahim et al. investigated the effects of PI3K inhibition in BRCA-proficient TNBC’s preclinical models with PI3K-activating alterations [147]. They observed that PI3K blockade induces (HR) impairment and sensitization to PARP inhibition [147]. Batalini et al. have recently published the results of a phase 1b trial showing that alpelisib, in combination with olaparib, has antitumor activity in patients with pre-treated TNBC [148]. An additional clinical study investigating buparlisib is currently ongoing [149]. Results from the aforementioned trials will provide new insights into the efficacy of this combination, further promoting the use of PI3Ki as an emerging therapeutic strategy in TNBC.

#### 3.4.4. Immunotherapy

The tumor microenvironment (TME) plays an essential role in tumor initiation, growth, invasion, metastasis and cancer treatment [24,150]. Specifically, TME allows cancer cells to become invasive and spread from the primary site to distant locations through a complex and multistep metastatic process [27]. Recently, the PI3K/AKT pathway has been shown to exert a pivotal role in regulating anti-tumor immunity by promoting an immunosuppressive TME and controlling the activity of the tumor infiltration cells associated [151]. Multiple studies have demonstrated how programmed death ligand-1 (PD-L1) and cytotoxic T lymphocyte-associated protein 4 (CTLA-4) interact with PI3K signalling. For instance, PI3K inhibition led to a reduction of tumor PD-L1 expression in PTEN-mutant TNBC and colorectal cancer (CRC) [152]. More specifically, the PI3Kα-specific or pan-PI3K inhibitor did not show an anti-tumor response over ICI alone in TNBC models, while the PI3K/mTOR dual inhibitor gedatolisib associated with ICIs induced a substantial cancer growth inhibition and a greater activation and response of T-cells, natural killer (NK)-cell, and dendritic cells (DC) [153].

On the other hand, in PTEN loss melanoma, preclinical and clinical studies have provided strong evidence that PI3Kβ inhibition, in combination with anti-CTLA-4 agent, improved the efficacy of immunotherapy [154,155]. Likewise, Lastwika et al. have demonstrated that in human lung adenocarcinomas and squamous cell carcinomas, PD-L1 expression was significantly correlated with mTOR activation [156]. Their findings were corroborated by studies using genetically engineered mouse models of lung cancer where an mTOR inhibitor, combined with a PD-1 antibody, reduced tumor growth, increased tumor-infiltrating T cells and diminished regulatory T cells [156].

At ASCO 2021, Schmid reported the preliminary results of the phase1b BEGONIA trial, evaluating the safety and efficacy of capivasertib with paclitaxel and durvalumab as a first-line treatment for PD-L1+ metastatic TNBC [124]. The addition of capivasertib resulted in an ORR similar to the paclitaxel/durvalumab arm, although the limited number of patients enrolled in the study does not allow robust conclusions to be drawn. Furthermore, the addition of capivasertib to durvalumab and paclitaxel regimen induced a relatively high rate of G3/4 treatment-related adverse events [121]. A phase 2 trial (NCT03961698) has investigated the triplet combination of eganelisib (PI3K-γ inhibitor) with atezolizumab and nab-paclitaxel as first-line therapy for locally advanced or metastatic TNBC patients and renal cell carcinoma (MARIO-3 trial) [122]. At the last update, this combination provided manageable toxicity and evidence of a long-term PFS benefit, in TNBC, with an ORR of 55.3% irrespective of PD-L1 expression [123]. Regarding ongoing clinical trials, a phase I/II trial (NCT03131908) is investigating the selective PI3K-beta inhibitor GSK2636771 in combination with pembrolizumab in patients with refractory metastatic PTEN-loss melanoma. Additionally, a phase I/II trial (NCT04317105) is evaluating copanlisib with nivolumab and ipilimumab in PI3K/AKT-mutated solid tumors. Another phase 2 trial (NCT03190174) is investigating the biological activity of the sequential administration of nivolumab and the mTOR inhibitor ABI-009 in multiple types of cancer. Another phase 1 trial (NCT03772561) is exploring capivasertib combined with durvalumab and olaparib in patients with advanced or metastatic solid tumors. Finally, a phase Ib study investigating the anti-PD1 antibody spartalizumab plus everolimus in TNBC patients (NCT02890069) has recently closed, although the outcome is not yet available. Recently, preclinical data have shown that PI3Ki may increase the efficacy of chimeric antigen receptor T cells (CAR-T) in vivo; however, these results are preliminary and further investigation is required to elucidate the underlying mechanism [157,158]. Several combination approaches with the PI3K/AKT/mTOR inhibitor and ICI are ongoing (Table 4).

## 4. Impact of Biomarkers

As mentioned above, differents oncogenic genomic alterations are responsible for PI3K/AKT/mTOR pathway hyperactivation. Somatic point mutations and gene amplifications are the two principal alterations promoting the PIK3CA functions [159]. PIK3CA gene status can vary among primary tumor and metastases [160]. This potential discordance can interest the gain or loss of the PIK3CA gene mutations or different levels of mutation [161,162]. All these aspects underline the importance of molecular characterization of metastatic sites on the activity of PI3Kis. The use of circulating tumor DNA (ctDNA) is an alternative when the biopsy of a metastatic site is difficult or cannot be obtained, selecting patients with adequate tumor burden or with disease progression to increase the probability of adequate ctDNA [163,164]. Beyond PIK3CA mutations, additional biomarkers have been evaluated as potential biomarkers of resistance to PI3Ki [165]. For instance, preclinical data showed that tumors characterized by PTEN expression loss are more sensitive to AKT/PI3K inhibitors and more dependent on PI3Kβ signalling, thereby benefiting from pictilisib [166,167]. On the other hand, the increase of insulin level induced by PI3Kis can lead to the re-activation of PI3K/AKT in murine tumour models [168], while PI3K inhibition may upregulate ER-dependent transcription due to the epigenomic crosstalk between PI3K and ER pathways [169].

Indeed, in a phase 1b trial of alpelisib and letrozole, the small subgroup of patients harbouring FGFR1/2 amplification, KRAS or TP53 mutations did not show any benefit [169]. It is important to underline that only PIK3CA mutation is currently approved as a predictive biomarker in clinical practice. Moreover, PIK3CA mutations have been recently included in the tier IA of genomic alterations in BC, of the ESMO Scale for Clinical Actionability of molecular Targets (ESCAT), as predictors of benefit from a-selective PI3Kis.

Ultimately, the PI3K pathway is involved in the differentiation of MDSCs and Tregs within TME, and suppressive cytokines can impair stimulation of macrophages and dendritic cells, suggesting a potential synergy for combining PI3Kis and immunotherapy [7,8]. The use of programmed PD-L1 and tumor-infiltrating lymphocytes (TILs) as predictive and prognostic biomarkers is well-recognized and associated with response to immunotherapy, although the latter biomarker is related to pathological specimens often found in the primary tumor site [170,171]. Several biomarkers of immunological state such as circulating tumor cells, circulating immunity cells and inflammatory indexes have been investigated as predictive and prognostic biomarkers [172,173,174,175].

Looking to the future, trials are being developed using baseline, on-treatment and post-treatment PI3Kis and immunotherapy which may improve our understanding of the complex interaction between host immunity and PI3KCA, ultimately improving our approach to patients.

## 5. Summary and Conclusions

Dysregulation of PI3K/AKT/mTOR signalling is frequently observed in human cancer and it is responsible for tumorigenesis, cancer progression, as well as intrinsic and acquired resistance to several treatments. This pathway is an attractive molecular target for therapeutic interventions and the development of novel anti-cancer molecules. The last two decades have seen exponential growth in the number of PI3K inhibitors investigated in pre-clinical studies, with approximately fifteen compounds that have progressed into clinical trials as new anticancer drugs. However, the high toxicity and the lack of selectivity have hampered the application and approval of PI3Ki in clinical practice. Clinical adverse events associated with these PI3K/AKT/mTORi such as hyperglycemia, pneumonitis, stomatitis, rashes and diarrhoea have a crucial impact on a patient’s quality of life leading to a high percentage of treatment discontinuation.

In order to optimize the efficacy of PI3Ki and limit toxicity, better management of side effects as well as well-designed studies for the identification and validation of actionable predictive biomarkers associated with the clinical activity are required.

Furthermore, additional investigations to establish the role of the PI3K pathway on the tumor microenvironment and a better patient selection and stratification will be crucial to pave the way for combination treatments with immunotherapies.

Due to their potential synergistic action, drug combination strategies with chemo or target therapies as well as novel dosing schedules may enhance the clinical benefit and potentially overcome intrinsic and acquired resistance with fewer AEs. In order to maximise combination therapies, high-throughput molecular profiling approaches will be essential to promote an accurate matching of patients with PIK3CA aberrations to specific tumor subtypes.

## Figures and Tables

**Figure 1 cancers-15-00703-f001:**
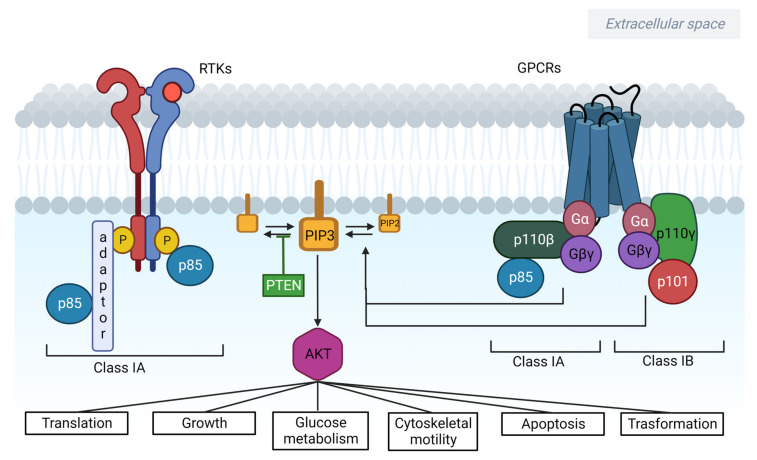
The PI3K/AKT/mTOR pathway is involved in tumorigenesis and cancer progression. After being activated by RTKs, GPCR or RAS, PI3K catalyzes the phosphorylation of PIP2 to generate PIP3, which binds and recruits AKT and PDK1. Furthermore, by activating NF-κB and inducing the secretion of MMP, AKT promotes cell invasion while increasing the level of cyclin D1, leading to cell cycle progression. Ultimately, Akt promotes cell growth by phosphorylation of the downstream mTORC1, which activates p70S6K-S6 and inhibits 4E-BP1, resulting in protein synthesis and cell growth. Indeed, mTORC2 activates AKT itself. On the other hand, PTEN exerts its role in modulating the PI3K pathway by suppressing PIP2 to PIP3 conversion. Together with tuberous sclerosis protein 1 (TSC1) and TSC2, PTEN is the main negative regulator of the pathway. Simultaneously, activation of the growth factor receptor tyrosine kinases and G protein-coupled receptors induces RAS/RAF/MEK/ERK signalling, and ERK activation can further contribute to mTORC1 activation.

**Figure 2 cancers-15-00703-f002:**
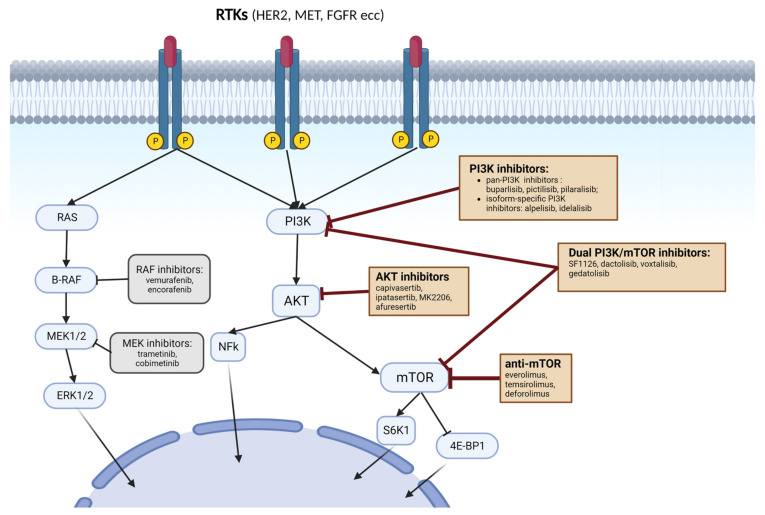
Summary of the complex phosphatidylinositol-3-kinase (PI3K)/AKT/mTOR signalling pathway and inhibitors.

**Table 1 cancers-15-00703-t001:** Summary of ongoing phases I–III trials with PI3k inhibitors in solid tumors.

Clinical Trial	Study Design	Intervention	Settings	Primary Endpoint	Phase	Status
NCT04975958	63 Participants Interventional Non-Randomized Parallel Assignment Open Label	Buparlisib Atezolizumab AN0025	Advanced Solid tumors	DLTs	1	Recruiting
NCT04338399 (BURAN)	483 Participants Interventional Randomized Parallel Assignment Open Label	Buparlisib Paclitaxel	mHNCC	OS	3	Recruiting
NCT04108858	12 Participants Interventional Parallel Assignment Open Label	Copanlisib Pertuzumab Trastuzumab	PI3KCA/PTEN mutated HER2+/HR- MBC	AEs	1/2	Recruiting
NCT04572763	48 Participants Interventional Non-Randomized Single-Group Assignment Open Label	Copanlisib Venetoclax	Relapsed/refractory DLBCL	MTD, ORR	1/2	Active Not recruiting
NCT03711058	18 Participants Interventional Non-Randomized Sequential Assignment Open Label	Copanlisib Nivolumab	MSS CRC	MTD, DLT	1/2	Active Not recruiting
NCT04253262	13 Participants Interventional Non-Randomized Sequential Assignment Open Label	Copanlisib Rucaparib	mCRPC	MTD	1/2	Active Not recruiting
NCT03502733	48 Participants Interventional Single-Group Assignment Open Label	Copanlisib Ipililumab Nivolumab	Advanced cancer, Lymphoma	RP2D	1	Active, Not recruiting
NCT03484819	106 Participants Interventional Single-Group Assignment Open Label	Copanlisib Hydrochlorid Nivolumab	Refractory DLBCL PMBCL	ORR	2	Active, Not recruiting
NCT02367040 CHRONOS-3	458 Participants Interventional Randomized Parallel Assignment	Copanlisib Rituximab	Relapsed iNHL	PFS	2	Active, Not recruiting
NCT01660451	227 Participants Interventional Non-Randomized Parallel Assignment Open Label	Copanlisib	Indolent or aggressive NHL	ORR		Active, Not recruiting
NCT05143229	18 Participants Interventional Non-Randomized Sequential Assignment Open Label	Alpelisib Sacituzumab Govitecan	Stage III/Stage IV HR+/HER2− MBC	RP2D	1	Recruiting
NCT04208178 (EPIK-B2)	551 Participants Interventional Randomized Parallel Assignment	Alpelisib Trastuzumab Pertuzumab	PIK3CA mutated HER2+ MBC	PFS	3	Recruiting
NCT04762979	44 Participants Interventional Single-Group Assignment Open Label	Alpelisib Fulvestrant Aromatase inhibitor	PIK3CA mutated HR+/HER2− MBC	PFS	2	Recruiting
NCT05508906	60 Participants Interventional Non-Randomized Parallel Assignment Open Label	Alpelisib Ribociclib OP-1250	HR+/HER2− MBC	DLTsMTD	1	Recruiting
NCT04251533	566 Participants Interventional Randomized Parallel Assignment	Alpelisib Nab paclitaxel Placebo	PIK3CA mutated/PTEN loss mTNBC	PFS, ORR	3	Recruiting
NCT05025735	25 Participants Interventional Randomized Single-Group Assignment Open Label	Alpelisib Dapagliflozin Fulvestrant	PI3KCA mutated HR+/HER2− MBC	Incidence of all grade hyperglycemia	2	Recruiting
NCT05230810	40 Participants Interventional Single-Group Assignment Open Label	Alpelisib Fulvestrant Tucatinib	PIK3CA mutated HER2+ MBC	AEs	1/2	Recruiting
NCT05501886 (VIKTORIA-1)	701 Participants Interventional Randomized Parallel Assignment Open Label	Alpelisib Gedatolisib Palbociclib Fulvestrant	HR+/HER2− MBC	PFS	3	Recruiting
NCT04997902	36 Participants Interventional Parallel Assignment Open Label	Alpelisib Tipifarnib	mHNCC	DLTs	1/2	Recruiting
NCT05063786	358 Participants Interventional Single-Group Assignment Open Label	Alpelisib Olaparib Paclitaxel PLD	metastatic OC	PFS	3	Recruiting
NCT04526470	55 Participants Interventional Single-Group Assignment Open Label	Alpelisib Paclitaxel	PIK3CA mutated GA	MTD RP2D	1/2	Recruiting
NCT03207529	28 Participants Interventional Single-Group Assignment Open Label	Alpelisib Enzalutamide	AR+/PTEN positive MBC	MTD	1	Recruiting
NCT01872260	255 Participants Interventional Randomized Parallel Assignment Open Label	Alpelisib Letrozole LEE011	HR+/HER2− MBC	DLTs Safety	1/2	Active, Not yet recruiting
NCT03284957 (AMEERA-1)	136 Participants Interventional Randomized Parallel Assignment Open Label	Amcenestrant Palbociclib Alpelisib Everolimus Abemaciclib	HR+/HER2− MBC	DLTs		Active, Not yet recruiting
NCT04666038 (BRUIN CLL-321)	250 Participants Interventional Randomized Parallel Assignment Open Label	Idelalisib LOXO-305 Bendamustine Rituximab	Chronic CLL/SLL	PFS	3	Recruiting
NCT03890289 (GAUDEALIS)	5 Participants Interventional Single-Group Assignment Open Label	Idelalisib Obinutuzumab	Refractory FL	ORR	2	Active Not yet recruiting
NCT02787369	3 Participants Interventional Non-Randomized Parallel Assignment Open Label	Idelalisib ACY-1215 Ibrutinib	Refractory CLL	MTD	1	Active Not yet recruiting
NCT02970318	311 Participants Interventional Randomized Parallel Assignment	Idelalisib calabrutinib (ACP-196) Rituximab Bendamustine	Refractory CLL	PFS	3	Active Not yet recruiting
NCT02135133	50 Participants Interventional Single-Group Assignment Open Label	Idelalisib Ofatumumab	CLL/SLL	ORR	2	Active, Not recruiting
NCT04191499	400 Participants Interventional Randomized Parallel Assignment	Inavolisib Palbociclib Fulvestrant	PIK3CA mutated HR+/HER2+ MBC	PFS	2/3	Recruiting

**Abbreviations:** AEs: adverse events; AR: androgen receptor; CLL: chronic lymphocytic leukemia; CRC: colorectal cancer; DLBCL: diffuse large B-cell lymphoma; DLTs: dose-limiting toxicities; mHNCC: metastatic head and neck cancer; MBC: metastatic breast cancer; iNHL: indolent B-cell non-Hodgkin’s lymphoma; FL: follicular lymphoma; mCRPC: metastatic castration-resistant prostate cancer; MSS: microsatellite stable; MTD: maximum tolerated dose; NHL: non-Hodgkin’s lymphomas; OC: ovarian cancer; ORR: objective response rate; OS: overall survival; PMBCL: primary mediastinal large B-cell lymphoma; PLD: pegylated liposomal doxorubicin; PFS: progression-free survival; RP2D: recommended phase 2 dose; SLL: small lymphocytic lymphoma.

**Table 2 cancers-15-00703-t002:** Summary of ongoing phases I–III trials with AKT-inhibitors in tumors.

Clinical Trial	Study Design	Intervention	Settings	Primary Endpoint	Phase	Status
NCT03310541	12 Participants Interventional Parallel Assignment Open Label	Capivasertib, Enzalutamide, Fulvestrant	Advanced solid tumors harboring mutations in AKT1, AKT2, or AKT3	ORR	1	Active, not yet recruiting
NCT05593497 (SNARE)	30 Participants Interventional Single-Group Assignment Open Label	Capivasertib, Abiraterone Acetate Leuprolide	PTEN loss High-risk localized PC	pCR MRD	2	Not recruiting
NCT04439123 (MATCH-Subprotocol Y)	35 Participants Interventional Single-Group Assignment Open Label	Capivasertib	Cancers with AKT genetic changes	ORR	2	Active, not yet recruiting
NCT04851613	20 Participants Interventional Non-Randomized Single-Group Assignment Open Label	Afuresertib, Fulvestrant	Locally advanced or HR+/HER2− MBC	ORR	1	Recruiting
NCT04374630 (PROFECTA-II)	141 Participants Interventional Parallel Assisgnment Open Label	Afuresertib Paclitaxel	Platinum-resistant ovarian cancer	rPFS	2	Recruiting
NCT05383482	167 Participants Non-Randomized Sequential Assignment Open Label	Afuresertib Nab paclitaxel Docetaxel Sintilimab	Solid tumors Resistant to prior anti-PD-1/PD-L1	AEs DLTs	1/2	Recruiting
NCT05390710	101 Participants Randomized Sequential Assignment Open Label	LAE005 + Afuresertib Nab-Paclitaxel	Metastatic TNBC	AEs DLT	1/2	Recruiting
NCT04060394	74 Participants Randomized Sequential Assignment Open Label	Afuresertib LAE001/prednisone +	mCRPC	rPFS	1/2	Recruiting
NCT04253561 (IPATHER)	25 Participants Interventional Single-Group Assignment Open Label	Ipatasertib Trastuzumab Pertuzumab	HER2+ PI3KCA mutant MBC	RP2D	1	Recruiting
NCT05172245	36 Participants Interventional Single-Group Assignment Open Label	Ipatasertib Cisplatin Radiation Therapy	Stage III-IVB HNC	MTD RP2D	1	Recruiting
NCT04467801 (Ipat-Lung)	60 Participants Interventional Single-Group Assignment Open Label	Ipatasertib Docetaxel	mNSCLC	PFS	2	Recruiting
NCT03673787	87 Participants Interventional Non-Randomized Parallel Assignment Open Label	Ipatasertib atezolizumab	Glioblastoma Multiforme mPC	MTD	1/2	Recruiting
NCT05276973	24 Participants Interventional Single-Group Assignment Open Label	Ipatasertib Carboplatin Paclitaxel	Stage III or IV Epithelial OC	MTD	1	Recruiting
NCT03959891 (TAKTIC)	60 Participants Interventional Non-Randomized Parallel Assignment Open Label	Ipatasertib Fulvestrant Aromatase Inhibitor Palbociclib	HR+/HER2− mBC	TEAE	1	Recruiting
NCT04650581 (FINER)	250 Participants Interventional Randomized Parallel Assignment	Ipatasertib Fulvestrant	HR+/HER2− mBC	PFS	3	Recruiting
NCT04920708 Without ctDNA Suppression (FAIM)	324 Participants Interventional Randomized Parallel Assignment Open Label	Ipatasertib Fulvestrant Palbociclib	HR+/HER2− mBC	PFS	2	Not yet recruiting
NCT04464174 (PATHFINDER)	54 Participants Interventional Non-randomized Parallel Assignment Open Label	Ipatasertib non-taxane chemotherapy	mTNBC	Safety	2	Active, not yet recruiting
NCT03853707	28 Participants Interventional Randomized Parallel Assignment Open Label	Ipatasertib Atezolizumab Capecitabine Carboplatin Ipatasertib Paclitaxel	mTNBC	RP2D, PFS	1/2	Active, not yet recruiting
NCT05498896 (BARBICAN)	146 Participants Interventional Non-randomized Parallel Assignment Open Label	Ipatasertib Atezolizumab Paclitaxel Doxorubicin Cyclophosphamide	mTNBC	pCR	2	Active, not yet recruiting
NCT03072238 (IPATential150)	1101 Participants Interventional Parallel Assignment	Ipatasertib Abiraterone Placebo	mCRPC	rPFS	3	Active, not yet recruiting
NCT05538897	96 Participants Interventional Randomized Parallel Assignment Open Label	Ipatasertib Megestrol Acetate	mEC	AEs	1/2	Not yet recruiting
NCT04739202 ((IMMUNOGAST)	60 Participants Interventional Non-Randomized Parallel Assignment Open Label	Ipatasertib Atezolizumab	mGA	ORR	2	Recruiting

**Abbreviations:** AEs: adverse events; DLT: dose-limiting toxicities; HNC: head and neck cancer; mBC: metastatic breast cancer; mCRPC: metastatic castration-resistant prostate cancer; mEC: metastatic endometrial cancer; mGA: metastatic gastric adenocarcinoma; MRD: minimal residual disease; mPC: metastatic prostate cancer; MTD: maximum tolerated dose; ORR: objective response rate; OC: ovarian cancer; PC: prostate cancer; pCR: pathologic complete response; PFS: progression free survival; rPFS: radiographic progression-free survival; RP2D: recommended phase 2 dose; TEAE: treatment-emergent adverse events.

**Table 3 cancers-15-00703-t003:** Summary of ongoing phases II-III trials with dual PI3K/mTOR inhibitors.

Clinical Trial	Study Design	Intervention	Settings	Primary Endpoint	Phase	Status
NCT03698383	15 Participants Interventional Single-Group Assignment Open Label	Trastuzumab biosimilars (Herzuma) Gedatolisib	HER2+ MBC	ORR	2	Recruiting
NCT03911973	52 Participants Interventional Single-Group Assignment Open Label	Talazoparib Gedatolisib	mTNBC	ORR	½	Recruiting
NCT03065062	96 Participants Interventional Single-Group Assignment Open Label	Palbociclib Gedatolisib	Solid tumors	MTD, RP2D	1	Recruiting
NCT05501886 (VIKTORIA-1)	141 Participants Interventional Randomized Parallel Assisgnment Open Label	Palbociclib Fulvestrant Alpelisib Gedatolisib	HR+/HER2 MBC	PFS	3	Recruiting

**Abbreviations**: MBC: metastatic breast cancer; MTD: maximum tolerated dose; ORR: objective response rate; PFS: progression-free survival; RP2D: recommended phase 2 dose (RP2D).

**Table 4 cancers-15-00703-t004:** Ongoing clinical trials combining immune checkpoint inhibitors with inhibitors of the PI3K/AKT/mTOR pathway.

Clinical Trial	Study Design	Intervention	Settings	Primary Endpoint	Phase	Status
NCT04431635	35 Participants Interventional Non-Randomized Single-Group Assignment Open Label	Copanlisib Nivolumab Rituximab	Relapsed/refractory indolent follicular or marginal zone lymphoma	MDT CR rate	Ib	Recruiting
NCT03961698	91 Participants Interventional Non-Randomized Parallel Assignment	Eganelisib Atezolizumab Nab-paclitaxel Bevacizumab	Metastatic TNBC or advanced RCC	CR rate	II	Active, not recruiting
NCT04317105	102 Participants Interventional Non-Randomized Parallel Assignment Open Label	Copanlisib Ipilimumab Nivolumab	Advanced malignant solid neoplasm	AEs, DLT	I/II	Recruiting
NCT03131908	36 Participants Interventional Non-Randomized Parallel Assignment Open Label	GSK2636771 Pembrolizumab	Metastatic PTEN loss melanoma	MTD ORR	I/II	Active, not recruiting
NCT03772561	40 Participants Interventional Non-Randomized Single-Group Assignment Open Label	Capivasertib Olaparib Durvalumab	Advanced or metastatic solid tumor malignancies	ORR	I	Recruiting
NCT05387616	98 Participants Interventional Non-Randomized Single-Group Assignment Open Label	Copanlisib Obinutuzumab	Follicular lymphoma	PFS	II	Recruiting
NCT03711058	54 Participants Interventional Non-Randomized Sequential Assignment Open Label	Copanlisib Nivolumab	MSS relapsed/refractory solid tumors and CRC	DLTs, ORR	I/II	Active, not recruiting
NCT03673787	87 Participants Non-Randomized Parallel Assignment Open Label	Ipatasertib Atezolizumab	Advanced solid tumours with PI3K pathway hyperactivation	MTD, AEs	I/II	Recruiting
NCT02637531	219 Participants Interventional Non-Randomized Single-Group Assignment Open Label	Eganelisib Nivolumab	Advanced solid tumours	DLTs, AEs	I/Ib	Active, not recruiting

**Abbreviations:** AEs: adverse events; CLL: chronic lymphocytic leukemia; CR: complete response rate; CRC: colorectal cancer; DLTs: dose-limiting toxicities; TNBC: triple negative breast cancer; MSS: microsatellite stability; MTD: maximum tolerated dose; ORR: objective response rate; PFS: progression-free survival; RCC: renal cell carcinoma.

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
