# Peer review of "Current State and Future Challenges for PI3K Inhibitors in Cancer Therapy"

_cancers, 2023, doi:10.3390/cancers15030703_

Round 1
Reviewer 1 Report
Sirico et al have submitted a comprehensive review of the PI3K/AKT/mTOR pathway in cancer with a particular focus on drugs in development and trials in progress. There is a detailed overview of components of the pathway followed by discussion of the drugs targeting these pathway components, and overall the authors summarize the field nicely. As this is a very broad topic with limited space available for this review, not every relevant discussion point can be detailed. However I would make the following suggestions:
1. In the introduction to Section 2, I would recommend mentioning PIK3R1 mutation as a mechanism for PI3K pathway activation in cancer.
2. Section 4 of "Impact on biomarkers" - this section focuses specifically on PIK3CA, but many other gene alterations activate the PI3K pathway in cancers, and prediction of clinical benefit based on genetic, gene expression and IHC biomarkers is of particular interest to the audience. Some additional discussion around these biomarkers, particularly as they relate to therapeutic vulnerabilities conferred by PTEN loss, would strengthen this review.
There are some grammatical, formatting and spelling errors in this review that hopefully will be corrected in editing, but I want to call attention to the following:
Ipatasertib is misspelled as "ipasertib" in section 3.2
Gedatolisib is misspelled as "getasolisib" in 3.4.2 and in Figure 2.
In Figure 2, pan-PI3K inhibitors is misspelled as "inhibotrs"
In Figure 2, the arrows from dual PI3K/mTOR inhibitors should point to PI3K and mTOR, not PI3K and AKT as shown
Capivasertib is misspelled as "capivartesib" in 3.5.1
In section 4: "Somatic point mutations and gene amplifications are the two principal alterations 2 impairing the PIK3CA functions" impairing should be "promoting" or "impacting"
Author Response
"Please see the attachment"

Reviewer 2 Report
A timely review article by Dr. Sirico and the group elaborates on the role of current PI3K inhibitors in clinical aspects and their translational aspect in treatment regimens. It's a well-written review article that discusses ongoing clinical trials for respective drug regimens. However, a few things need to be addressed before it is accepted. They are as follows:
1. Authors need to discuss the differential dosage of rapamycin affecting mTOR activity differentially. This topic has been well described in PMID: 26916116 and PMID: 24508508. The authors should add a few lines on this topic.
2. It has been discussed well how RAS/MAPK pathways interact with PI3K/mTOR signaling. While it's a well-known fact that PI3K pathways play a definite role in cancer metabolism (PMID: 31686003), it has also been discussed recently how oncogenic RAS/ MAPK pathways play a role in cancer metabolism (PMID: 33870211). So it will be worthwhile to discuss as a potential field of study to determine the metabolic changes upon the combination of RAS/MAPK pathway inhibitor and PI3K/mTOR inhibitions. This will be a fascinating field of study in the near future. Authors need to discuss this topic by adding a few lines to this.
3. In figure 2, please add the names of BRAF, and MEK inhibitors, as mentioned for AKT and PI3K inhibitors.
Round 2
Reviewer 1 Report
The authors appropriately responded to my review and I have no further comments.
Reviewer 2 Report
All concerns have been addressed, ready for acceptance.